# Associations of water contact frequency, duration, and activities with schistosome infection risk: A systematic review and meta-analysis

**Fabian Reitzug, Julia Ledien, Goylette F. Chami** *

Big Data Institute, Nuffield Department of Population Health, University of Oxford, Oxford, United Kingdom

* goylette.chami@ndph.ox.ac.uk

## Abstract

### Background

Schistosomiasis is a water-borne parasitic disease which affects over 230 million people globally. The relationship between contact with open freshwater bodies and the likelihood of schistosome infection remains poorly quantified despite its importance for understanding transmission and parametrising transmission models.

### Methods

We conducted a systematic review to estimate the average effect of water contact duration, frequency, and activities on schistosome infection likelihood. We searched Embase, MED-LINE (including PubMed), Global Health, Global Index Medicus, Web of Science, and the Cochrane Central Register of Controlled Trials from inception until May 13, 2022. Observational and interventional studies reporting odds ratios (OR), hazard ratios (HR), or sufficient information to reconstruct effect sizes on individual-level associations between water contact and infection with any *Schistosoma* species were eligible for inclusion. Random-effects meta-analysis with inverse variance weighting was used to calculate pooled ORs and 95% confidence intervals (CIs).

### Results

We screened 1,411 studies and included 101 studies which represented 192,691 participants across Africa, Asia, and South America. Included studies mostly reported on water contact activities (69%; 70/101) and having any water contact (33%; 33/101). Ninety-six percent of studies (97/101) used surveys to measure exposure. A meta-analysis of 33 studies showed that individuals with water contact were 3.14 times more likely to be infected (OR 3.14; 95% CI: 2.08–4.75) when compared to individuals with no water contact. Subgroup analyses showed that the positive association of water contact with infection was significantly weaker in children compared to studies which included adults and children (OR 1.67; 95% CI: 1.04–2.69 vs. OR 4.24; 95% CI: 2.59–6.97). An association of water contact with

**Data Availability Statement:** All relevant data are within the manuscript and supplementary files. 

**Funding:** Scholarship from the Nuffield Department of Population Health to FR. Funding

from the Wellcome Trust Institutional Strategic Support Fund (204826/Z/16/Z), Nuffield Department of Population Health Pump Priming Fund, John Fell Fund as part of the SchistoTrack Project, a Robertson Foundation Fellowship, and UKRI EPSRC Award (EP/X021793/1) all to GFC. The funders had no role in study design, data collection and analysis, decision to publish, or preparation of the manuscript.

**Competing interests:** The authors have declared that no competing interests exist.

infection was only found in communities with $\geq$10% schistosome prevalence. Overall heterogeneity was substantial ($I^2$ = 93%) and remained high across all subgroups, except in direct observation studies ($I^2$ range = 44%–98%). We did not find that occupational water contact such as fishing and agriculture (OR 2.57; 95% CI: 1.89–3.51) conferred a significantly higher risk of schistosome infection compared to recreational water contact (OR 2.13; 95% CI: 1.75–2.60) or domestic water contact (OR 1.91; 95% CI: 1.47–2.48). Higher duration or frequency of water contact did not significantly modify infection likelihood. Study quality across analyses was largely moderate or poor.

## Conclusions

Any current water contact was robustly associated with schistosome infection status, and this relationship held across adults and children, and schistosomiasis-endemic areas with prevalence greater than 10%. Substantial gaps remain in published studies for understanding interactions of water contact with age and gender, and the influence of these interactions for infection likelihood. As such, more empirical studies are needed to accurately parametrise exposure in transmission models. Our results imply the need for population-wide treatment and prevention strategies in endemic settings as exposure within these communities was not confined to currently prioritised high-risk groups such as fishing populations.

## Author summary

Schistosomiasis is a neglected tropical disease endemic to 78 countries and affects over 230 million people globally, mostly in sub-Saharan Africa. Contact with open freshwater sources such as lakes, rivers, and dams puts people at risk of infection. Existing systematic reviews have linked schistosome infection to risk factors such as gender and a lack of access to clean drinking water and hygiene infrastructure. Yet, while these risk factors may be mediated by water contact behaviours, the direct association of water contact with schistosome infection likelihood remains poorly quantified. We conducted a systematic review of associations between water contact frequency, duration, and activities on schistosome infection likelihood. We found that having any water contact was associated with 3.14 times higher likelihood of infection compared to having no water contact. This relationship held across adults and children and across medium and high-prevalence settings. Out of 11 water contact activities, 10 actives were associated with increased infection risk. Occupational, domestic, and recreational exposure was associated with similar infection likelihood. The evidence on exposure-schistosome infection associations included in this meta-analysis comes largely from self-reported measures elicited through cross-sectional surveys. Over 90% of included studies did not mention measurement of water contact as one of their aims, meaning exposure measurement was not their primary focus. Study quality was mostly moderate or low and recall periods employed in studies varied between one day and one year. There is a need for an expert consensus to consistently measure exposure. Studies should report on the measurement periods and definitions of water contact categories that were used. Few studies reported on correlations of water contact with infection intensity; this information is needed from future studies to better understand transmission.

## Background

Human schistosomiasis is a chronic water-borne parasitic disease which affects over 230 million people and is endemic to 78 countries across Asia, the Middle East, North Africa, South America, and particularly sub-Saharan Africa [1–3]. The causative agents of schistosomiasis, trematodes of the genus *Schistosoma*, are transmitted through contact with open freshwater sources. Infection occurs when the free-swimming stage of the parasite penetrates a human host by boring through the skin. Local transmission depends on human water contact behaviour and environmental conditions including climate and the characteristics of waterbodies and their suitability for the parasite's intermediate host; competent freshwater snail species [4,5]. While it is known that the high focality of schistosomiasis is due to the setting-specific interplay between humans and the environment, the factors determining local prevalence are complex and remain incompletely understood [4,6].

Water contact behaviours in endemic areas, such as swimming, fishing, bathing, washing, or laundry play a key role in determining infection likelihood and are highly variable across small spatial scales. Here, exposure is defined as having water contact in sites with schistosome cercaria. There is variation in water contact duration, frequency, and activities within and across households and villages, across different age groups, genders, and occupational groups [7–13]. Mathematical models have shown that communities with more heterogeneous levels of water contact, as measured by water contact frequency, have higher levels of transmission compared to communities with homogenous water contact, all else being equal [14]. Therefore, knowledge of water contact behaviour is important to understand community transmission and individual-level likelihood of schistosome infection.

The magnitude of associations between water contact and schistosome infection has varied across studies, and some studies did not find any significant association. Costa et al. [15] conducted a survey in rural south-eastern Brazil and found that water contact was positively associated with increased odds of schistosome infection in people aged 15 years and older (OR 6.8; 95% CI: 3.5–13.2). The association was significantly stronger in people of age 14 years and younger (OR 55.8; 95% CI: 27.2–114.6). By contrast, a nationally representative cross-sectional survey in Uganda by Exum et al. [16] found that self-reported contact with open freshwater sources was not significantly associated with schistosome infection, even when the effect size was unadjusted (prevalence ratio 1.08; 95% CI: 0.9–1.26).

Past studies have used surveys, diaries, direct observation, and wearable global positioning system (GPS) loggers to measure water contact. Surveys provide cross-sectional measures of current exposure. Self-reported measures have been found to agree relatively poorly with observation-based measures of water contact in studies which compared the two [17,18]. Direct water contact observation using trained local observers has provided longitudinal measures of exposure within communities. This data has contributed to understanding typical levels of exposure in endemic settings as well as daily, weekly, monthly, and seasonal water contact patterns. Observation studies have been able to document large numbers of water contacts over extended time periods. For instance, Fulford et al. conducted water contact observations in seven Kenyan communities spanning three years and recording 67,000 contacts [9]. Sow et al. observed water contacts in a community in northern Senegal for two years, recording 120,000 water contacts [11]. The accuracy of water contact observation has been validated by assessing inter-observer agreement [9,11]. Limitations of observation are that it is expensive, data processing time consuming, and that resulting exposure measures have often been used as ecological, not individual-level, measures [9,11,19–21]. Among the two above-mentioned observation studies [9,11], none reported on individual-level infection outcomes. A method that has been made possible by advances in wearable technology (lower cost, smaller

size, longer battery life) [22] are water contact measurements using GPS loggers. GPS loggers provide longitudinal measures but so far, the existing studies in Cameroon and Uganda have been restricted to short observation periods (24 hours and 72 hours, respectively) and small samples (n = 24, n = 97) [17,23]. Across all water contact measurement tools, no accepted gold standard for exposure measurement exists.

There is a need to systematically analyse associations between water contact and schistosome infection to 1) disentangle the relative role of exposure versus acquired immunity in explaining individuals' likelihood of infection [24–27], 2) assess whether there is a dose response-relationship between degree of exposure and schistosome infection intensity [10,11,13], and 3) examine the extent of the age-dependency of infection [27–30]. It remains still an open question as to what water contact measures are associated with schistosome infection status and intensity. Therefore, we performed a systematic review and meta-analysis to address the following question. What is the average effect of water contact frequency, duration, and activity type on the likelihood of schistosome infection?

## Methods

### Protocol and search strategy

The protocol was registered on May 13, 2022 with PROSPERO (CRD42022333680) [31]. The systematic review is reported in accordance with the Preferred Reporting Items for Systematic Reviews and Meta-Analyses (PRISMA) guidelines [32]. The completed checklist is available in S1 Text.

We searched six databases in English on May 13, 2022 (Embase (1974-), MEDLINE (including PubMed, 1946-), Global Health (1973-), Global Index Medicus (1901-), Web of Science Core Collection (1900-), Science Citation Index Expanded (1900-), and the Cochrane Central Register of Controlled Trials (CENTRAL, 1996-). The following search string was used "Schistosomiasis" AND ("water" (AND "contact" OR "pattern"OR "duration"OR "frequency" OR "behaviour"OR "exposure")) AND ("risk" OR "infection" OR "intensity" OR "transmission OR "odds" OR "likelihood"). We used a combination of free text search and Medical Subject Heading terms. S2 Text details the full search strategy.

### Exposures and outcome

The Population-Exposure-Comparators-Outcomes (PECO) framework was used to formulate the research question [33]. The study population were people of any age or gender living in areas endemic with a species of schistosome. Water contact was the exposure of interest and was grouped into four distinct categories: having any water contact, water contact duration, water contact frequency, and water contact activities. Having any water contact was defined as a binary measure of water contact, i.e., whether the respondent had any water contact with an open freshwater body over the specified recall or observation period. Frequency of water contact was defined as a categorical measure indicating the number of times per day, week, or month that the respondent had contact with an open freshwater body. Duration of water contact was defined as a categorical measure, indicating the duration of water contact per exposure event or per time period of recall or observation. Water contact activities were defined as binary measures of whether the respondent engaged in a given water contact activity (e.g., bathing, swimming, fishing). Water contact measures from any exposure measurement tool were eligible. The comparators were individuals without any water contact or with low water contact frequency or low duration, dependent on the unit of exposure. The primary outcome measure was schistosome infection (either infection status or intensity) based on a recognised diagnostic such as microscopy (e.g., Kato-Katz method [34], urine filtration [35]) or antigen or

antibody-based methods (e.g. point-of-care circulating cathodic antigen test [36,37] or polymerase chain reaction-based detection [38]). Effect measures were odds ratios or hazard ratios with 95% confidence intervals (CIs). Wherever available, we extracted hazard ratios or adjusted odds ratios. Otherwise, unadjusted odds ratios (ORs), or information to convert reported associations to ORs, such as risk ratios or 2x2 tables were used.

## Inclusion and exclusion criteria

Eligible study designs included cohort studies, cross-sectional studies, case-control studies, before-after studies (if reporting on baseline infection and exposure) and randomised controlled trials (if reporting on infection and exposure at baseline or in the control group). We excluded studies that used self-reports or (micro-)haematuria to ascertain infection status. Only exposure measures with a clearly identifiable dimension of water contact, such as having any water contact, water contact frequency, duration, or activity categories, were included. Exposure indices that collapsed multiple different dimensions of water contact (and sometimes also measures of environmental risk) into a single metric and which could not be mapped onto the four water contact dimensions were excluded.

## Screening and data extraction

Title, abstract, and full-text screening was performed independently by two reviewers (FR and JL). Data was extracted by FR. A randomly selected sample of 10% of all included studies was independently extracted by JL for quality control. When studies contained duplicated data because they reported on the same population, the study with the larger sample size was retained. If the data to reconstruct measures of association were not reported, associations could not be reconstructed from published data including supplements or required information was not provided by the authors after e-mail contact, the study was excluded.

We used a structured, pre-tested form and extracted the following study characteristics: study design, study location, study setting, local waterbodies, sampling strategy, participant characteristics, inclusion/exclusion criteria, exposure measurements, *Schistosoma* species, definition of the schistosome infection outcome, diagnostic test, and schistosome infection prevalence in study participants. More details on the data extraction are available in S3 Text. References of all included studies and a list of excluded studies can be found in S1 Table and S2 Table, respectively.

## Subgroup variables

We grouped studies by various characteristics for subgroup analyses. Children were defined as ages 0–14 (pre-school-age children (PSACs) were ages 0–4, school-age children (SACs) were ages 5–14). Adults were age groups 15 and older. Studies were grouped by endemicity setting following the 2022 World Health Organization (WHO) guideline definitions for mass drug administration (MDA) treatment (low < 10% prevalence, moderate = 10–49% prevalence, high ≥ 50% prevalence). We grouped waterbodies by transmission-relevant characteristics– stagnant versus flowing waterbodies and artificial versus natural waterbodies—which have been found to affect risk in past studies [2,39]. Stagnant waterbodies were defined as canals, dams/reservoirs, lakes, ponds, fields whereas streams and rivers were flowing. Artificial waterbodies were canals, dams/reservoirs whereas all other waterbodies were defined as natural. We assigned the local climate zone for each study using the GPS study location and the Köppen-Geiger climate classification [40] (S3 Text). For the duration meta-analysis, we defined two exposure categories: water contact duration ≤ one hour, duration > one hour. Water contact frequency was grouped into three categories: daily, weekly, and monthly water contact. For the

activity meta-analysis, water contact activities were grouped into three broader categories (domestic, recreational, occupational water contact). S3 Table provides details on how we grouped water contact categories.

## Data analysis

For all possible exposure-outcome pairs, random-effects meta-analyses with inverse variance weighting following the DerSimonian and Laird approach were conducted whenever at least three studies reported comparable effect measures [41]. We preferentially used adjusted effect sizes because they account for potential confounding factors in the exposure-outcome relationship. In the water contact duration and activity analyses where it was possible for studies to contribute more than one effect size, we used multilevel meta-analysis. Wherever possible, we performed pre-specified subgroup analyses. Again, we only show groups representing at least three studies. Subgroup analyses could not be performed for age, gender, length of exposure measurement period, and time elapsed between exposure and outcome assessment due to insufficient numbers of studies reporting required information. Between-study heterogeneity assessments were conducted using the $I^2$-statistic, Egger's test, and funnel plots. All models were implemented using the 'meta' and 'dmetar' packages in R version 4.2.1 [42–44].

## Risk of bias

The quality of all studies was appraised. A modified version of the Quality Assessment Tool for Observational Cohort and Cross-Sectional Studies from the National Institutes of Health (NIH, Bethesda, MD, United States of America) was used for this purpose and risk of bias was divided into three categories (low, moderate, and high, S4 Text) [45]. Our preferred method for assessing the effect of study quality on the results was to perform meta-analyses separated by risk of bias. As few studies were rated high quality, we did not have enough studies to use this method in frequency, duration, and activity meta-analyses. In this case, we resorted to influence analysis as our secondary method and removed one study at a time from each meta-analysis to quantify the effect of any single study on the results.

# Results

## Study characteristics

The search returned 1,411 unique titles and abstracts. 495 studies met the inclusion criteria for full text review; 101 studies were included (Fig 1). Summary characteristics of all included studies are reported in Table 1. Detailed study-level information and full references of all included studies are provided in S1 Table. S2 Table provides a list of excluded studies and details their exclusion reasons. Of the 101 studies in this systematic review representing 192,691 participants and reporting on 501 associations of water contact with schistosome infection, 98 studies (97%) with 191,146 participants provided comparable outcome and exposure measures and were included in at least one full or subgroup meta-analysis.

Ten studies (10%) explicitly mentioned measuring human water contact as one of their study aims. The remaining 91 studies (90%) had a different primary focus but still reported on water contact. Studies included in the systematic review were from 22 different countries across three continents (Africa, Asia, South America, Fig 2A). With 10–13 studies from each country, Brazil, Ethiopia, Nigeria, and Cameroon represented approximately 50% of all included studies. The earliest included study was from 1984 with most studies after 2010 (Fig 2B). The most frequently studied water settings were as follows: 58 studies (57%) reported on rivers, 35 studies (35%) on streams, 24 studies (24%) on lakes, and 21 studies (21%)

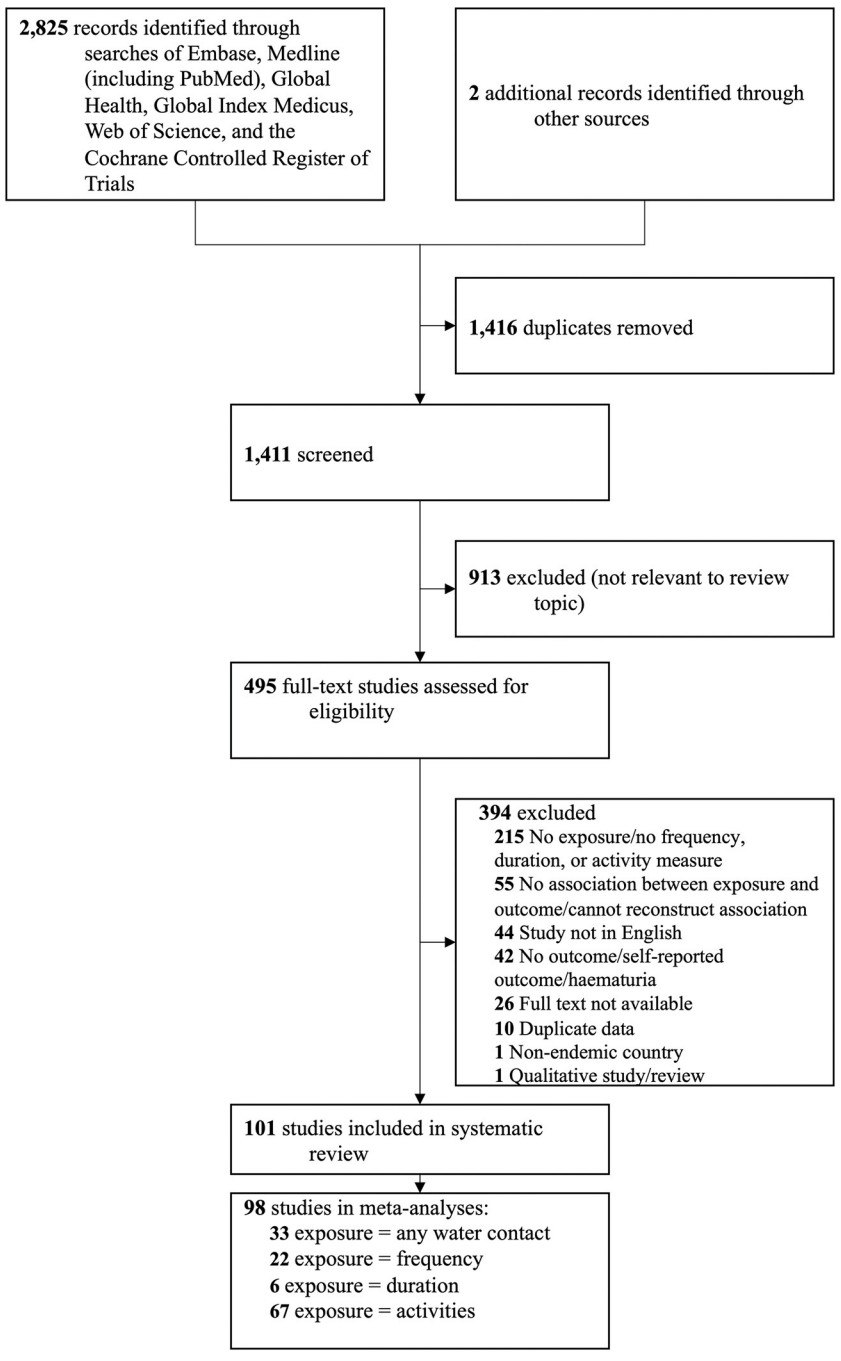

**Fig 1. Study selection.** PRISMA Flowchart depicting the number of records identified, included, and excluded, and the reasons for exclusion.

reported on ponds. Thirty-six studies (36%) mentioned multiple types of waterbodies and 65 studies (65%) mentioned a single type of waterbody.

Studies screened at the full-text stage used four different water contact measurement tools: surveys, direct water contact observation using trained human observers, participant water contact diaries, and GPS loggers. No diary study and no GPS logger study provided required ORs or sufficient information to reconstruct ORs. Of the 101 included studies, 98 studies

**Table 1. Summary of study characteristics.**

| Characteristics | N = 101 |
|---|---|
| **Study aims** | |
| measurement of water contact | 10 (9.9%) |
| other aim | 91 (90.1%) |
| **Continent** | |
| Africa | 78 (77.2%) |
| Asia | 10 (9.9%) |
| South America | 13 (12.9%) |
| **Locality** | |
| rural | 77 (76.2%) |
| urban | 4 (4.0%) |
| mixed[a] | 20 (19.8%) |
| **Study setting** | |
| community | 63 (62.4%) |
| school | 31 (30.7%) |
| other[b] | 7 (6.9%) |
| **Population[c]** | |
| PSACs/SACs | 33 (32.7%) |
| PSACs/SACs/adults | 68 (67.3%) |
| ***Schistosoma* species** | |
| *S. haematobium* | 47 (46.5%) |
| *S. mansoni* | 42 (41.6%) |
| *S. japonicum* | 5 (5.0%) |
| multiple species | 7 (6.9%) |
| **Endemicity** | |
| low (<10% prevalence) | 14 (13.9%) |
| medium (10–49% prevalence) | 65 (64.4%) |
| high (≥50% prevalence) | 22 (21.8%) |
| **Köppen-Geiger climate zone** | |
| arid, desert, hot | 13 (12.9%) |
| arid, steppe, hot | 6 (5.9%) |
| temperate, dry summer, warm summer | 1 (1.0%) |
| temperate, dry winter, hot summer | 3 (3.0%) |
| temperate, dry winter, warm summer | 4 (4.0%) |
| temperate, no dry season, hot summer | 10 (9.9%) |
| tropical, monsoon | 11 (10.9%) |
| tropical, rainforest | 3 (3.0%) |
| tropical, savannah | 50 (49.5%) |
| **Water setting** | |
| river | 58 (57.4%) |
| stream | 35 (34.7%) |
| lake | 24 (23.8%) |
| pond | 21 (20.8%) |
| dam/reservoir | 10 (9.9%) |
| canal | 10 (9.9%) |
| field | 3 (3.0%) |
| swamp | 2 (2.0%) |
| **Number of waterbodies studied** | |

(*Continued*)

**Table 1.** (Continued)

| Characteristics | N = 101 |
|---|---|
| single | 65 (64.4%) |
| multiple[d] | 36 (35.6%) |
| **Nature of waterbody** | |
| artificial waterbody | 19 (18.8%) |
| natural waterbody | 82 (81.2%) |
| **Flow of waterbody** | |
| flowing waterbody | 46 (45.5%) |
| stagnant waterbody | 55 (54.5%) |
| **Measurement tool** | |
| direct water contact observation | 4 (4.0%) |
| survey | 97 (96.0%) |
| **Water contact dimension** | |
| any water contact | 33 (32.7%) |
| frequency | 27 (26.7%) |
| duration | 6 (5.9%) |
| activity | 70 (69.3%) |
| **Study design** | |
| case-control | 4 (4.0%) |
| cohort | 10 (9.9%) |
| cross-sectional | 86 (85.1%) |
| randomised controlled trial | 1 (1.0%) |
| **Study quality** | |
| low | 46 (45.5%) |
| moderate | 49 (48.5%) |
| high | 6 (5.9%) |

[a]Locality ('mixed') = study areas that include both rural and (peri-)urban areas.

[b]Study setting ('other') = health clinic or multiple settings.

[c]Population ('PSACs') = pre-school-age children. Population ('SACs') = school-age children.

[d]Number of waterbodies ('multiple') = multiple different water settings included in the study (e.g. river and lake).

Full references of all included studies are provided in S1 Table.

(97%) used surveys and four studies (4%) used direct water contact observation. Water contact activities were reported in 70 studies (69%), followed by any water contact which was reported in 33 studies (33%). Twenty-seven studies (27%) reported on water contact frequency and 6 studies (6%) reported on duration. All activity measures came from surveys. Three of the four direct observation studies recorded whether people had any water contact and two recorded information on duration, and one recorded frequency.

## Association of any water contact with infection status

There were 33 studies which reported on associations between any water contact and schistosome infection status. Included studies represented 36,065 participants of whom 9,839 were infected with a schistosome species. The meta-analysis showed that having any water contact was associated with 3.14 times higher odds of infection (95% CI: 2.12–4.65) when compared to no water contact (Fig 3). There was no significant difference between studies reporting adjusted effect sizes versus studies reporting unadjusted effects but only 9 out of 33 studies provided any adjusted effect sizes (S1 Fig). When separating studies by reinfection versus

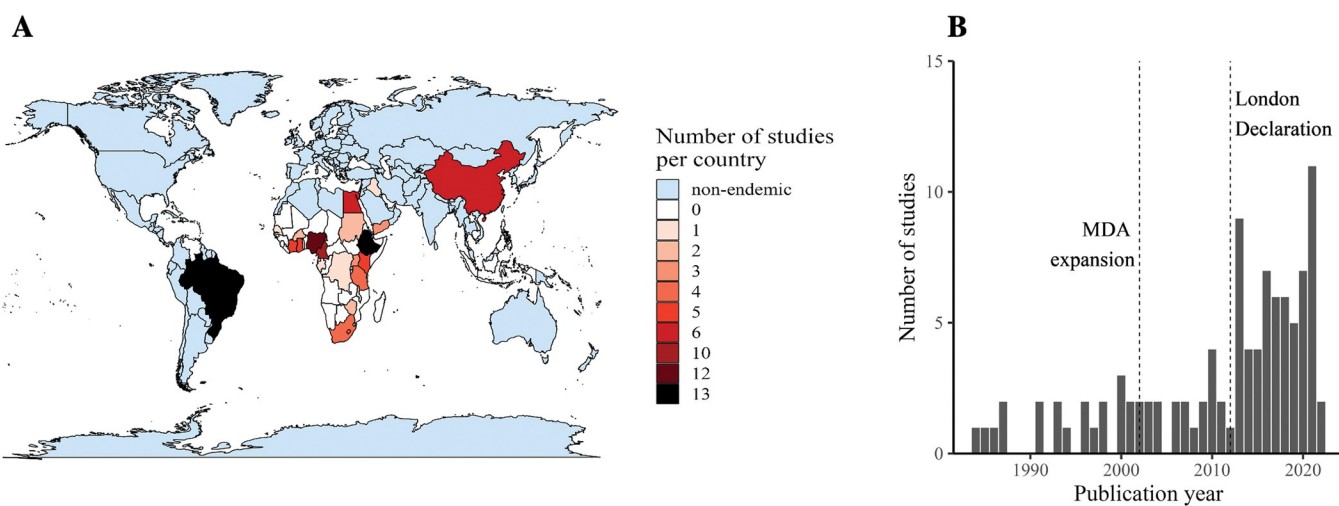

**Fig 2. Distribution of studies across geography and time.**

infection as the outcome, the effect on reinfection was not significant, but the reinfection subgroup was small and included only three studies (S2 Fig). When stratified by study quality (grouped into low, medium, high quality), pooled associations remained significant in all three strata and there were no significant differences between groups (S3 Fig). We did not find evidence for publication bias (Egger's test = 0.06, funnel plot provided in Fig 4). A post-hoc power analysis showed that this meta-analysis had over 99% power to detect the pooled effect of 3.14 we observed (S4 Fig). The meta-analysis was robust to removing any one study at a time from the analysis (S5 Fig). Study quality was rated as high in five studies (15%), 14 studies (42%) were rated as moderate, and 14 (42%) as low quality (see S4 Table for quality appraisal results).

Subgroup analyses in Fig 5 showed that associations of any water contact with infection were significantly different in studies including only PSACs and SACs compared to studies which included all age groups. Among PSACs and SACs, having water contact was associated with 1.67 times higher infection likelihood which was lower than the association in studies across all age groups (PSACs, SACs, and adults combined) where water contact conferred 4.24 times greater infection likelihood ($\chi^2$ = 7.06 df = 1, p<0.01). To understand the contribution of school children to this result, we separated studies by setting (S6 Fig) and found that water contact was significantly associated with infection in school-based studies (i.e., studies that included school children) but not in community-based studies (which included both SACs and PSACs). The exclusion of all school-based studies (S7 Fig) did not change the finding (Fig 5) that water contact was less strongly associated with infection in children compared to studies including children and adults. There was no overall difference of the effect of water contact on infection between community-based and school-based studies (Fig 5).

We found that community prevalence, type of water body, and climate zone modified the influence of water contact. In moderate endemicity settings having water contact was associated with 3.04 times higher likelihood of infection (95% CI: 1.88–4.89). In high endemicity settings the respective OR was 4.93 (95% CI: 1.99–12.22). There was no significant association between water contact and odds of infection in low endemicity settings. Among the waterbodies studied here water contact with streams and lakes but not with rivers was associated with schistosome infection likelihood. Reporting contact with streams was associated with 8.16 times higher infection likelihood (95% CI: 3.08–21.61) compared to no water contact. Contact

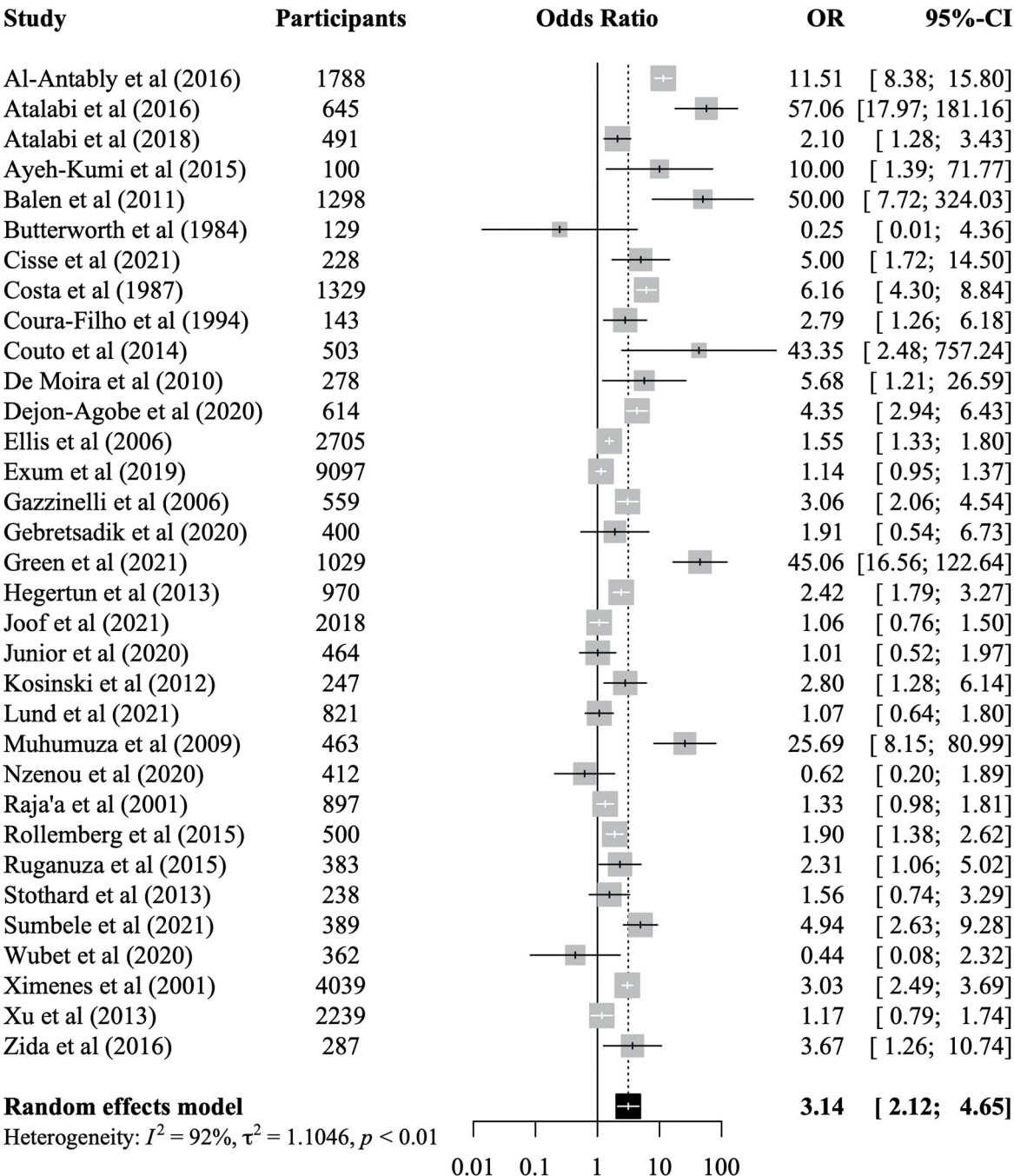

**Fig 3. Meta-analysis of the association between individual-level water contact and odds of schistosome infection.** Pooled effect sizes are based on random-effects models using data from 33 studies representing 36,065 participants. OR = odds ratio. CI = confidence interval. $I^2$ = I-squared statistic of heterogeneity.

with lakes was associated with 3.20 times higher risk of infection risk (95% CI: 1.09–9.37). The subgroup analysis in Fig 5 using the Köppen-Geiger climate zones showed that among the tropical, arid, and temperate zones represented here, only water contact in tropical savannah climate which is characterised by a pronounced dry season was significantly associated with infection likelihood (OR 3.35; 95% CI: 2.22–5.06). This climate zone is found among studies

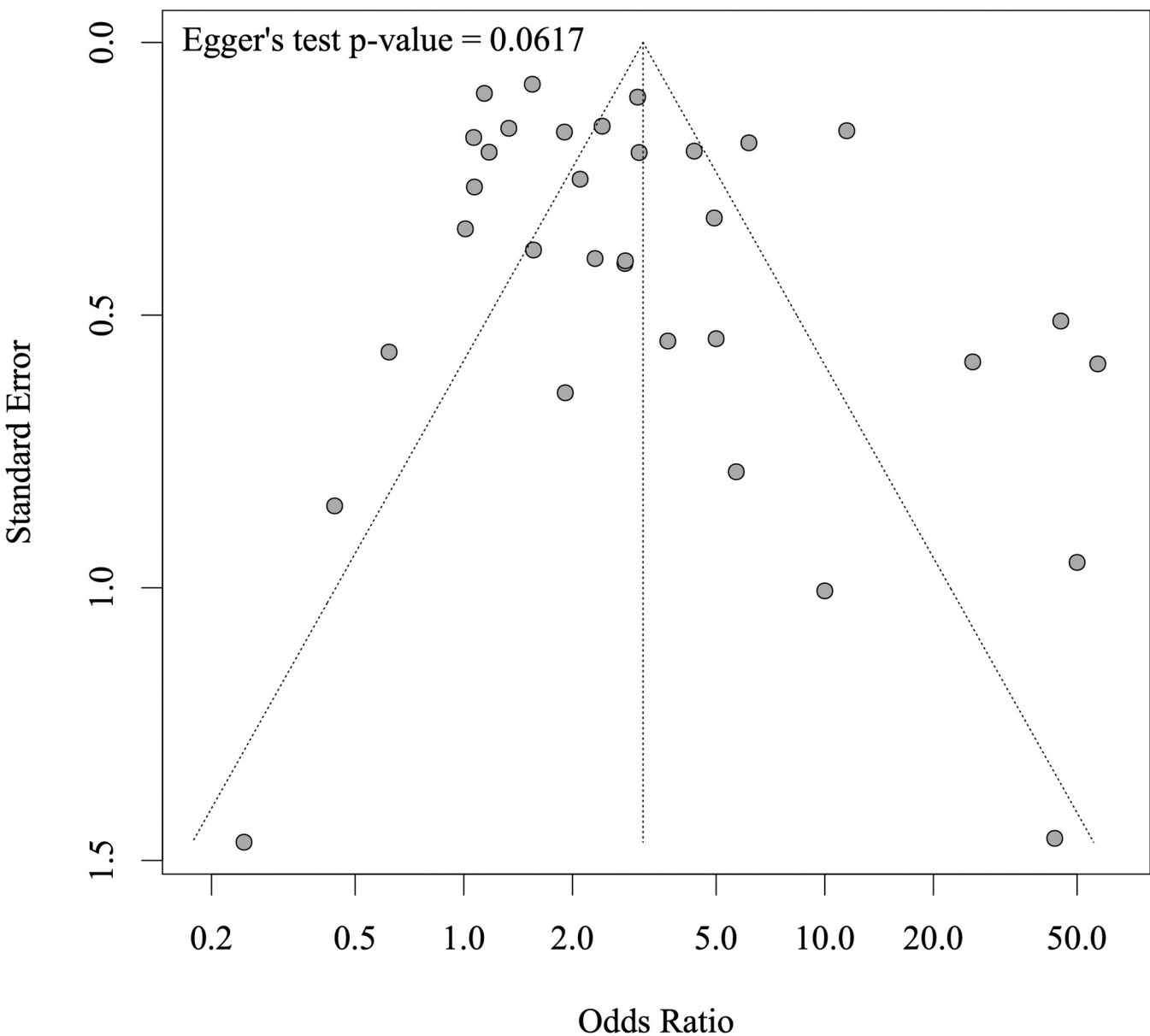

**Fig 4. Funnel plot for meta-analysis of the association between individual-level water contact and odds of schistosome infection.**

from sub-Saharan Africa and Brazil. The schistosome species present (*S. haematobium* versus *S. mansoni*) did not significantly modify infection likelihood. Descriptive data on the distribution of schistosome species across different waterbodies among the included studies suggested that *S. haematobium* was more frequently studied in research that focused on rivers or streams and *S. mansoni* was mostly the focus of studies which took place along lakes. Associations of water contact and schistosome infection were significant in cross-sectional studies (OR 3.36; 95% CI: 2.19–5.16) but not in cohort studies (OR 1.81; 95% CI: 0.76–4.33). Subgroup analyses also highlighted several study characteristics which did not significantly modify associations between water contact and infection likelihood. Associations remained significant across all types of localities (rural and mixed localities), artificial waterbodies and natural waterbodies and stagnant and flowing waterbodies. Associations in Fig 5 were also similar across

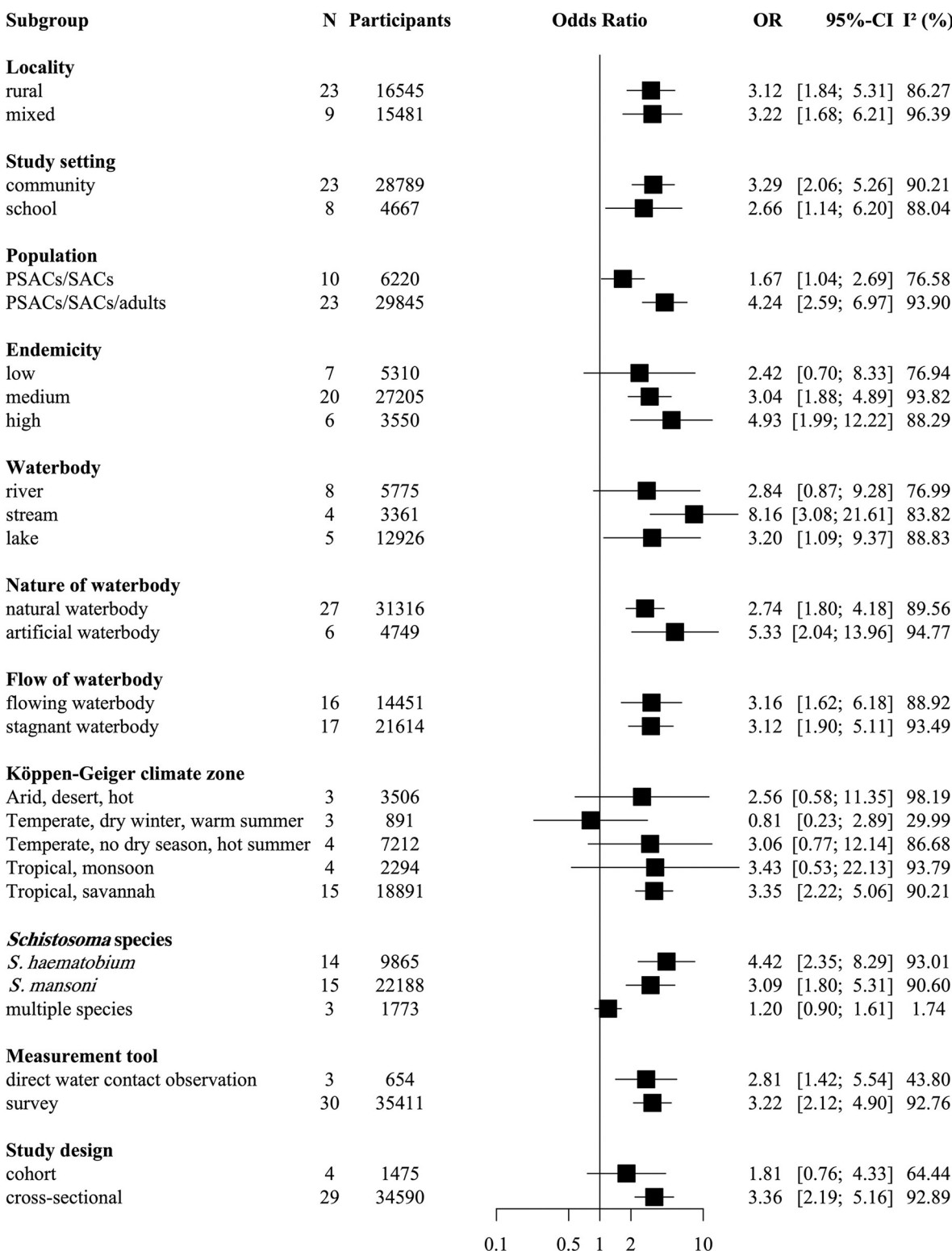

**Fig 5. Subgroup meta-analysis of the association between individual-level water contact and odds of schistosome infection.** Random-effects models using data from 33 studies representing 36,065 participants. N = number of studies. Categories with N<3 were not reported. OR = odds ratio. CI = confidence interval. $I^2$ = I-squared statistic of heterogeneity. Locality = 'mixed': study settings which include urban/peri-urban areas. PSAC = pre-school-age children. SAC = school-age children. Endemicity settings low/medium high = WHO definitions (low<10% prevalence, moderate = 10–49% prevalence, high≥50% prevalence). Artificial waterbody = canal, dam/reservoir. Stagnant waterbody = canal, dam/reservoir, lake, pond, field. Köppen-Geiger climate zone = climate zone at the study location.

measurement tools (surveys (OR 3.22; 95% CI: 2.12–4.90) and direct water contact observation (OR 2.81; 95% CI: 1.42–5.54)). Yet, heterogeneity was substantially lower when water contact was measured through direct observation ($I^2$-statistic of 44% versus 93% for surveys).

### Association of water contact frequency and duration with infection status

For water contact frequency, we meta-analysed 22 studies representing 19,208 participants of whom 5,957 were infected (Fig 6). Five studies could not be grouped into these categories as they either had unspecific frequency categories such as 'sporadic water contact', they reported frequency only on a subgroup, or they only reported on infection intensity. Any frequency of water contact (daily, weekly, monthly water contact) was associated with increased odds of infection when compared to no or lower levels of water contact. Although the magnitude of the ORs increased with more frequent water contact, these subgroup differences were not significant ($\chi^2$ = 1.69, df = 2, p = 0.43). Heterogeneity across subgroups was related to water contact frequency and decreased from daily ($I^2$ = 91%), weekly ($I^2$ = 80%) to monthly water contact ($I^2$ = 47%). Funnel plots and Egger's tests provided no evidence for publication bias (S8 Fig). Influence analyses in S9 Fig showed that the analysis was robust to removing any single study from the analysis. In this analysis, we rated study quality as high in three studies (14%), as moderate in ten studies (45%), and as low in nine studies (41%).

The meta-analysis of water contact duration was based on six studies representing 6,742 participants of whom 1,504 were infected (Fig 7). None of the water contact duration categories ($\leq$ 1 hour, > 1 hour) nor the overall pooled effect size for any duration was significantly associated with schistosome infection likelihood when compared to no water contact or lower levels of water contact. Due to the small number of studies in the duration meta-analysis (N = 6), no Egger's test and no influence analysis were conducted, and the funnel plot did not provide evidence for asymmetry (S10 Fig). The study quality was rated as moderate in four studies (67%), and as low in two studies (33%).

### Association of water contact activities with infection status

Activities were grouped into distinct categories as described in S3 Table. Sixty-seven studies representing 102,149 participants of whom 27,356 were infected, were included in the multilevel meta-analysis. Three studies reported on activity categories that could not be grouped. In the meta-analysis shown in Fig 8, 10 out of 11 activities were significantly associated with schistosome infection. These activities were crossing, urination, getting water, washing, swimming, playing, bathing, fishing, farming, and irrigation. Sand extraction was the only activity not associated with increased infection risk. Except for crossing and irrigation, the 95% CIs of all activities were overlapping. In this analysis, three of the included studies (4%) were rated as

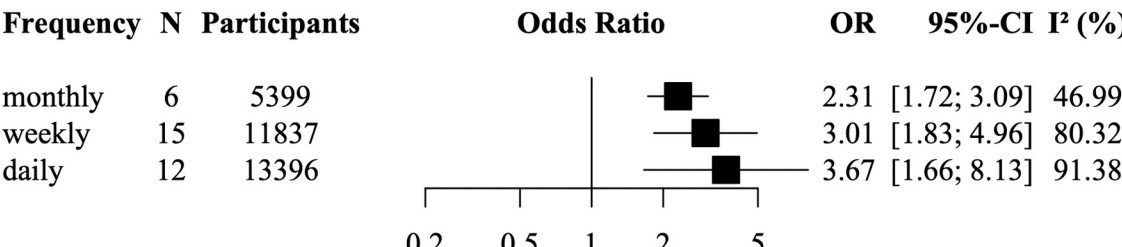

**Fig 6. Meta-analysis of the association between daily, weekly, monthly water contact and odds of schistosome infection.**
Random-effects models using data from 22 studies representing 19,208 participants. N = number of studies. OR = odds ratio. CI = confidence interval. $I^2$ = I-squared statistic of heterogeneity.

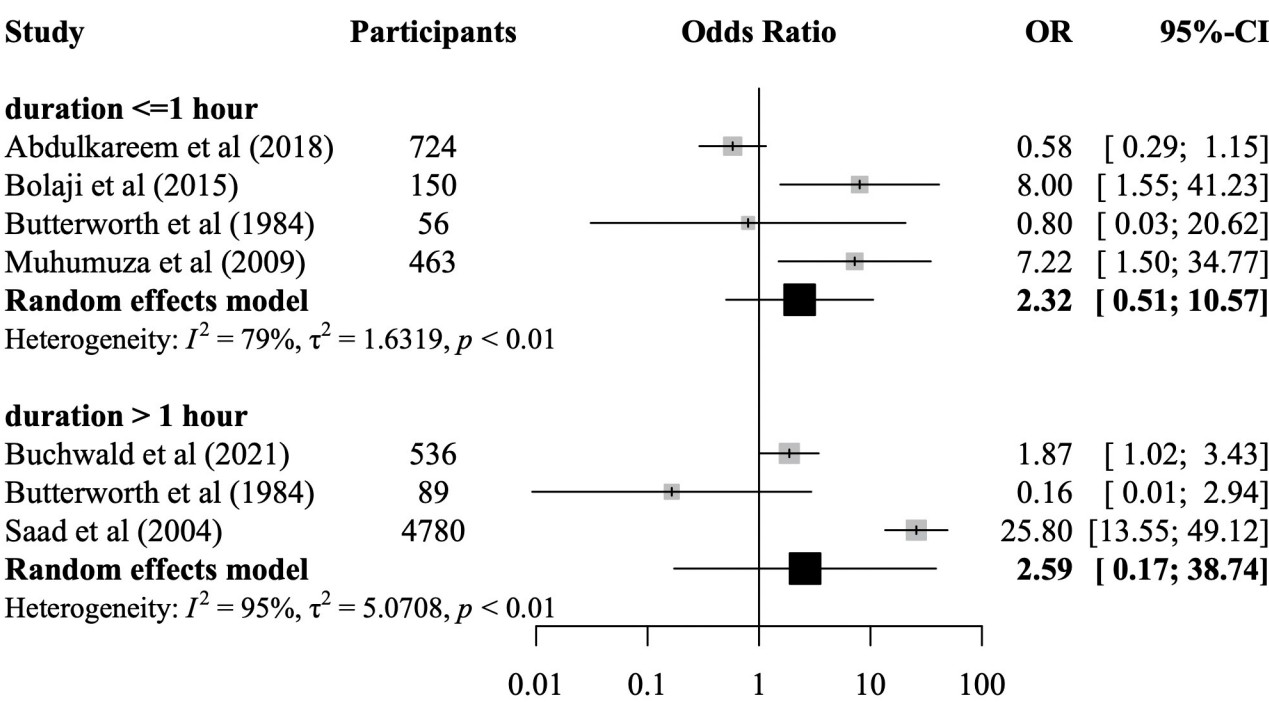

**Fig 7. Meta-analysis of the association between individual-level water contact duration and odds of schistosome infection.** Random-effects models using data from six studies representing 6,742 participants. OR = odds ratio. CI = confidence interval. $I^2$ = I-squared statistic of heterogeneity.

high quality, 36 (54%) as moderate, and 28 studies (42%) as low quality. When regrouping the activities into broader activity categories (domestic, recreational, or occupational water contact, see S3 Table), occupational activities had the highest point estimate for the odds of schistosome infection (OR 2.57) followed by recreational activities (OR 2.13) and domestic activities (OR 1.89), though none were significantly different from each other. Heterogeneity was high ($I^2$ of ≥90% in most categories) and Egger's test and funnel plots indicated risk of publication bias (S11 Fig). Influence analyses showed that almost all activity categories (except for sand extraction and urination) were robust to removing any single study from the analysis (S12 Fig).

## Discussion

Water contact behaviours play a key role in transmission but remain poorly understood and quantified [46]. Since the adoption of World Health Assembly 54.19 resolution in 2001 [47], MDA using praziquantel has become the cornerstone for schistosomiasis control. However, there is a need to complement MDA with WASH interventions and reductions in water contact to lower schistosomiasis prevalence [48]. To understand the role of exposure for transmission, we conducted a systematic review and meta-analysis of associations of water contact with schistosome infection and pooled data from 98 studies representing 191,146 participants across three continents. A simple binary measure of current water contact was strongly correlated to current infection status across studies.

### Any water contact

The magnitude of the association between exposure and schistosome infection suggests that water contact is a more influential risk factor for infection than WASH or gender. We found

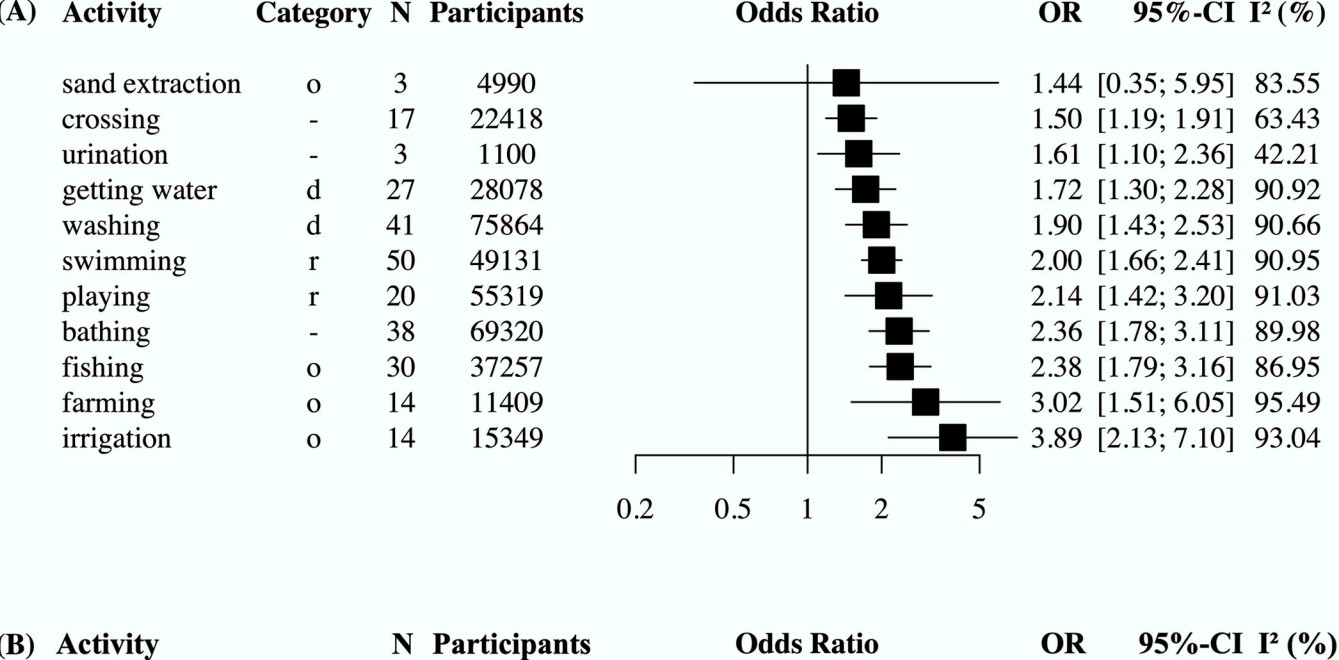

**Fig 8. Meta-analysis of the associations between water contact activities and odds of schistosome infection.** Panel (A): multilevel random-effects models using data from 67 studies representing 102,149 participants reporting on water contact activities. Washing = washing laundry, animals, clothes, utensils, dishes or blankets. Farming = farming, agriculture, rice farming. Irrigation = irrigation, watering gardens or cleaning streams. In the category column, o = occupational, d = domestic, r = recreational, '-' denotes activities that could not be grouped into these categories. Panel (B): multilevel random-effects models using data from 69 studies with 104,267 participants showing pooled effects sizes according to the water contact categories domestic, recreational, and occupational. N = number of effect sizes. Categories with N<3 were not reported. OR = odds ratio. CI = confidence interval. $I^2$ = I-squared statistic of heterogeneity.

that having any water contact was associated with 3.14 times higher odds of infection compared to no water contact. A meta-analysis by Ayabina et al. [49] found that schistosomiasis prevalence of males was 1.2 times the prevalence of females. The authors hypothesized that gender differences in water contact behaviour may explain their finding. Gender-disaggregated exposure data was rarely reported in our studies but is needed to investigate this hypothesis. A meta-analysis by Grimes et al. [50] showed that having no access to safe water was associated with 2.27 times higher odds of infection compared to having access to safe water supplies (estimate transformed from risk ratio to odds ratio for comparability). The relative magnitude of associations of infection status with WASH and with water contact, respectively, is plausible given that water contact is not fully determined by access to WASH. Even when safe water supplies are present, people may continue to have some degree of contact with open water sources [46].

This review shows that the understanding of the relationship between water contact and schistosome infection is mostly based on a single tool: cross-sectional surveys, most of which were not primarily concerned with studying water contact. Ninety-seven out of 101 studies (96%) were surveys. No diary or GPS logger studies were eligible for inclusion and only four

included studies used direct water contact observation. The dominance of surveys limited our ability to compare how associations varied across tools. Future research should focus on the quantitative analysis of water contact and infection using tools other than surveys to measure water contact. With more varied tools, conclusions could be drawn about the validity of different methods to measure exposure. We also found that only 10 included studies (10%) explicitly stated that their objective was to study the association between water contact and infection. Thus, the body of literature which mainly focuses on water contact is small compared to the number of studies which also reported on exposure but had a different primary study focus.

The association of current water contact with schistosome infection suggests that individual-level water contact behaviour might be stable across time. Present water contact cannot explain current infection status. This is because the time from infection to diagnosis is at minimum 4–6 weeks, depending on the diagnostic test used [51,52]. It has been acknowledged that present exposure is not necessarily representative of the relevant exposure history [53] because water contact varies over time [9,54]. Nonetheless, cross-sectional studies operate under the assumption that present water contact is an adequate proxy measure of current exposure. The same assumption is made in mathematical transmission models. For instance, the stratified-worm-burden model by Gurarie et al. uses cross-sectional exposure and infection data from survey for model calibration [55]. Our analysis found a robust association between having water contact and infection status which suggests that current exposure must be at least somewhat representative of past exposure. A similar conclusion has been reached by an observation study in Zimbabwe by Chandiwana and Woolhouse [53]. The authors interpreted associations between schistosome infection intensity and water contact over a two-week observation period as evidence for the persistence of exposure patterns over time. They noted that while there was seasonal variation in contact, individuals with high water contact tended to maintain high contact across seasons. Future studies with multiple timepoints of exposure measurement are needed to more fully investigate to which extent current and past contact are related.

The positive association between having any water contact and infection was significantly attenuated in studies that included only PSACs and SACs compared to studies which included PSACs, SACs, and adults. The weaker associations of water contact and infection in PSACs and SACs might seem surprising in light of numerous studies that found children to be at highest risk of infection [25,27,56,57]. The attenuated relationship in PSACs and SACs also remained when excluding all school-based studies. Among the 10 studies which sampled exclusively children, less than half of all studies (40%) were school-based. Therefore, it does not seem that the composition of the sample can explain why PSACs and SACs were at lower risk than all ages combined. As the difference between the group of PSACs and SACs and the group which represents all ages is that the latter includes adults, it is plausible that the addition of adults could explain the greater effect size in the combined group. However, due to a lack of studies which sampled adults only, we were unable to directly compare the risk of children versus adults and further investigate this hypothesis. The lack of studies reporting on age-disaggregated water contact does not allow any conclusions to be made on the shape of the exposure-infection relationship over age. Exposure and acquired immunity are well-established as explanations for the age dependency in schistosome infection, despite poorly established associations on dose-response relationships of water contact and infection likelihood [13,25–28]. To better study the contribution of water contact to the age dependency in infection, there is a need to sample adults in addition to SACs [58,59] and to report separately on their infection outcomes and water contact behaviours. We found no evidence that the likelihood of infection from water contact was any greater for PSACs and SACs compared to other age groups, suggesting that there is no support for children being more exposed to schistosome infection via water contact.

Past studies have found enrolled SACs to be at lower prevalence of infection compared to non-enrolled SACs [60–62]. We do not confirm this finding as SACs were more likely to be infected when sampled from schools when compared to PSAC and SACs sampled from communities. Limitations of this finding are that we did not have community-based studies reporting only on SACs and that there were relatively few studies (four school-based and five community-based studies) that allowed for comparisons between SACs sampled in schools versus PSACs and SACs sampled from communities.

## Water contact frequency and duration

This meta-analysis found no clear evidence that higher levels of exposure confer significantly higher schistosome infection risk compared to lower levels of exposure, as measured by water contact duration and frequency. It is plausible that relatively low levels of water contact are sufficient for becoming infected. A study by Moses et al. [63] in Uganda noted that water contact duration of as little as five minutes per day was significantly associated with *S. mansoni* infection risk. Therefore, water contact intensity may be less relevant for predicting individual-level infection status than knowledge on whether someone had any water contact [46]. Our results are consistent with findings from Moses et al. as even monthly water contact (the most infrequent exposure category) was significantly associated with the likelihood of infection. Another possibility is that our meta-analysis has been unable to establish the true relationship between the degree of exposure and infection status due to differential measurement error across exposure categories. The $I^2$ statistic of heterogeneity indicates that there is almost twice as much variability in the association of infection within the highest water contact frequency category compared to the lowest frequency category. The increase in within-subgroup heterogeneity with higher water contact frequency may indicate a challenge in reliably measuring more frequent water contact. This argument does not extend to duration where the $I^2$ for the pooled associations was similar between the two duration categories and in fact lower than for any of the other exposure measures. We were not able to conduct a meta-analysis of water contact frequency and schistosome infection intensity due to a lack of studies reporting on this outcome. However, most research on the dose-response relationship between schistosome infection and degree of exposure has focused on is infection intensity [10–12]. The dose-response relationship between exposure and infection intensity is especially relevant for transmission because heavily infected individuals may disproportionately affect local transmission [64,65]. Transmission dynamics have been incorporated into mathematical models to guide schistosomiasis control efforts, including those by the WHO [48]. The recently developed stochastic individual-based transmission model SCHISTOX uses an age-adjusted water contact frequency and a cercaria uptake rate to model exposure [66]. Yet, it is currently unclear how exposure should be parameterised in transmission models when, as we observed in this meta-analysis, few studies report on water contact and infection intensity.

## Water contact activities

Almost any type of water contact activity was linked to increased infection risk despite expected differences in activities regarding contact duration and degree of bodily immersion [4,11,29]. Our results contrast with a previous systematic review (without meta-analysis) by Grimes et al. which considered laundry, bathing and recreational swimming as the activities with most exposure and getting water as unimportant [46]. Our meta-analysis did not provide support for such a ranking of activities, surprisingly so as fishing has been a focus of many studies on schistosome risk [12,17,27,67,68]. The grouping of activities into high and low risk is implied by the 2006 WHO guidelines which define fishermen, farmers, irrigation workers,

and women executing domestic tasks as at-risk adult groups [69]. In our analysis, very few studies reported on schistosome infection likelihood for different at-risk groups. For instance, three studies (3%) reported on fishing as an occupation. At the same time, 29 studies (29%) reported on fishing as an activity but only six of them (21%) adjusted for any covariates, despite the availability of variables such as age and gender, which are well-known risk factors for infection [25,26,49]. Our analysis hence mostly included crude ORs for the infection likelihood of fishing. When grouping water contact activities into occupational, domestic, and recreational activities, we did not find that these categories conferred substantially different risks. To better understand the extent to which various activities are associated with different likelihoods of infection, future research needs to collect more detailed individual-level exposure information, not only on the type of activity, but also on frequency, duration, or degree of immersion of that activity. An insufficient number of included studies reported information that would have enabled subgroup meta-analyses of different activities, separated by frequency and duration.

## WASH thresholds and community infection prevalence

We found that having water contact in high prevalence areas ($\geq$ 50% prevalence) was associated with similar schistosome infection likelihood compared to water contact in medium prevalence areas (10–49%), using the 2022 WHO guideline definitions for MDA treatment [48]. To which extent community prevalence of schistosomiasis can be explained by water contact behaviour is still unclear. A study in Senegal of an immunological naïve population found that water contact levels were low and could not explain the very high community prevalence of schistosomiasis of 75%-100% [11]. By contrast, another study in Kenya found that persistent hotspot villages with poor response to annual MDA treatment differed from lower prevalence villages in their water contact patterns [70]. The hotspot villages had significantly higher proportions of individuals who engaged in bathing, washing clothes or dishes, getting water, playing, and swimming. Our finding that infection risk of water contact was similar across high-prevalence and medium-prevalence communities suggests that WASH, health education or behaviour change interventions that reduce water contact should not be targeted exclusively at high endemicity settings. In terms of WASH, our results suggest that achieving a critical threshold of (near-)universal community-level coverage may be needed for WASH to successfully control transmission. As almost any water contact activity was associated with infection risk, lowering community-level exposure and contamination may require not just access to safe drinking water, but providing a whole suite of facilities including laundry basins, showering or bathing facilities, toilets, or swimming facilities that minimise all types of open water contact. We were unable to directly assess how access to safe water or sanitation modified individual-level risk of water contact using subgroup analysis as none of the 33 studies in our main analysis provided effect sizes adjusted by access to safe drinking water or sanitation. For MDA, the implication is that treatment should extend beyond high endemicity settings and needs to reach all groups with regular water contact as the likelihood of (re)infection conferred by water contact was similar in high and medium endemicity settings.

## Waterbody types

This meta-analysis found that water contact with streams and lakes was significantly associated with likelihood of infection while contact with rivers was not. We found that artificial waterbodies (such as reservoirs, dams, canals) were associated with a similar likelihood of individual-level infection when compared to water contact with natural waterbodies. The result contrasts with a meta-analysis by Steinmann et al. [2] which found that prevalence in irrigated areas and

dammed landscapes was significantly higher when compared to pre-construction prevalence or prevalence in control areas. These differences could be due to different comparators. Steinmann et al. compare dams and irrigated areas (i.e., settings with favourable environmental conditions for transmission) against control areas with lower environmental risk while our review compared different types of waterbodies against each other, all of which may be high-risk settings. It is also unclear if differences in infection likelihood across water settings are driven primarily by environmental or behavioural factors. If the latter were the case, we would expect to see no difference in infection likelihood between water settings once we account for water contact behaviour. For example, the effect of dam construction on schistosomiasis could be driven by increased water contact through fishing or irrigation as environmental modification provides new economic opportunities. Such an occupational shift has been observed in Nigeria where fishing became an important water contact activity after dam construction [71]. More research is needed to understand the aquatic biology and behavioural patterns which shape infection outcomes associated with different waterbodies [4,6]. It is unclear if our finding of differential risk of different waterbodies (i.e., between rivers, lakes, and streams)–rather than reflecting substantial risk differences in these settings—may be spurious as some waterbodies are more well-studied than others. In our meta-analysis, more than half of all included articles studied streams, one third studied rivers, and a quarter studied lakes (multiple mentions were possible). By contrast, dams/reservoirs, canals, fields/rice paddies and swamps provided too few studies for subgroup analyses. Therefore, our meta-analysis had limited ability to contrast risk across the full range of water settings. Due to a lack of malacological data in most studies we could not investigate how the risk of water contact was mediated by the presence of intermediate snail hosts across different waterbodies. Among 101 studies, only 21.8% reported on snail abundance (from malacology) and 12.8% reported on prevalence of infected snails. The number of studies per subgroup was too low to analyse how risk of water contact varied between studies that found snails compared to studies that found no snails.

### Climate zones and seasonality

Seasonality modified associations between having water contact and schistosome infection. Having any water contact was associated with infection status only in study locations classified as tropical savannah climate by the Köppen–Geiger climate zone classification [40]. Tropical savannah climate is characterised by a pronounced dry season either in winter or summer. None of the other climate zones represented in the review had a similarly pronounced dry season. Seasonality has been shown to affect transmission both through changes in human behaviour with higher water contact in the dry season [9] and through variation in environmental conditions (such as water flow, rainfall, and temperature) that affect environmental risk [5,72]. Our finding is consistent with previous research that has linked dry-season water contact to elevated infection risk [54]. Moreover, our meta-analysis showed that within the tropical savannah studies, it was contact with seasonal (as opposed to perennial water sources) which dominated. Among included studies, 50% of waterbodies in tropical savannah climate were streams (a higher proportion than in other climates zones) and only 14% were rivers. Streams are more likely to be seasonal compared to rivers which lends further support to the hypothesis of seasonal water contact as a risk factor.

### Limitations

A limitation of this review was the high degree of heterogeneity ($I^2 > 90\%$). The $I^2$ value was comparable to heterogeneity in two other meta-analyses [49,50] of observational studies on risk factors for schistosomiasis infection whose $I^2$ statistics ranged between 82% and 96%.

Heterogeneity was lower only in the meta-analysis of associations between water contact duration and infection status where the $I^2$ was between 65% and 67%. None of the subgroup analyses conducted substantially reduced heterogeneity in the association between schistosome infection and having any water contact, except for the subgroup of direct observation studies which had half the degree of heterogeneity ($I^2 = 43\%$) compared to the subgroup of surveys ($I^2 = 93\%$). Egger's test indicated significant funnel plot asymmetry in the meta-analysis of water contact activities and infection. The source of this asymmetry is not entirely clear; it could reflect reporting bias, inflated effect sizes from smaller studies or true heterogeneity in associations [73].

Exposure measures may not have been entirely comparable across studies due to the absence of standardised survey instruments and varying recall periods and exposure definitions. Only one study mentioned the use of a standardised module to elicit self-reported water contact [74]. The recall periods employed in surveys varied widely between water contact in the past seven days to water contact in the past year and most studies failed to report their recall period. Some of the water contact activity categories may be overlapping and include somewhat heterogeneous measures, as reflected in the high $I^2$. There was an insufficient number of studies reporting on livestock farming, even though human-animal water contact is crucial especially for understanding risk for zoonotic transmission of schistosomiasis [75]. When grouping activities into the broader types (domestic, recreational, and occupational), the same participants may be included multiple times if they conducted several domestic activities of the same type, such as farming and fishing. However, we used multilevel models to account for repeated observations across activities within the same study. In addition, some dimensions of water contact that are relevant to infection likelihood were not represented in this review, such as bodily immersion and water contact by time of day [8,29].

## Conclusion

The review provides evidence that having water contact was robustly associated with infection in both children and adults, across different water and endemicity settings, and across measurement tools. The results highlight the key role that human exposure behaviour plays in explaining individual-level infection outcomes. The pooled effect size of the association is sizable compared to two meta-analyses of other risk factors; gender and lack of safe water access [49,50]. Our findings have implications for public health policies and interventions. Because all age groups, including adults, and people living in medium and high endemicity settings were at infection risk from water contact, there is a need for community-wide MDA and interventions to reduce exposure across all ages. This is in line with the 2022 WHO treatment guideline which recommends community-based treatment of all people aged two and above in communities with >10% prevalence. Fishing is often cited as a high-risk occupation, including in the 2022 WHO guidelines, but very few studies report directly on fishing as an occupation. Most evidence on fishing comes from studies reporting on fishing as an activity, but the majority did not adjust for any covariates. Studies on water contact and infection should include occupation as a covariate. There are several ways to improve measurement and reporting of water contact in studies on schistosomiasis. Survey-based studies should clearly state the recall periods and the survey questions which were used to elicit responses. The comparability of various duration and frequency measures across studies could be improved by using 'no water contact' as the reference category across analyses of frequency and duration. Similar to the UNICEF and WHO Joint Monitoring Programme definitions for WASH [76], it would be beneficial to create an expert consensus and assemble standardised tools for exposure measurement. Thus far, the literature has focused on the exposure measures of having any water

contact and the type of water contact activities. Future research needs to ascertain measures of frequency, duration, and immersion for different water contact activities. In addition, studies should routinely report on schistosome infection intensity as an outcome, not only infection status. To reduce schistosome transmission and develop more effective policies, more studies still are needed to inform accurate exposure measurements for ascertaining the age-dependency of exposure, identifying high-risk groups, and informing transmission modelling.

## Supporting information

**S1 Fig. Meta-analysis of the association between individual-level water contact and odds of schistosome infection–subgroup analysis of adjusted versus unadjusted effect sizes.** Full references of all included studies are available in S1 Table.
(JPEG)

**S2 Fig. Meta-analysis of the association between individual-level water contact and odds of schistosome infection–subgroup analysis of infection/reinfection.** Full references of all included studies are available in S1 Table.
(TIFF)

**S3 Fig. Meta-analysis of the association between individual-level water contact and odds of schistosome infection–subgroup analysis by study quality (low, moderate, high).** Full references of all included studies are available in S1 Table.
(TIFF)

**S4 Fig. Power analysis for meta-analysis of the association between individual-level water contact and odds of schistosome infection.** Post-hoc power calculation to detect an OR of 3.14 with 31 studies and a median study size of 503 participants >99% (parameters set to values from the meta-analysis).
(TIFF)

**S5 Fig. Influence analysis of any single study in the meta-analysis of the association between individual-level water contact and odds of schistosome infection.** Full references of all included studies are available in S1 Table.
(TIFF)

**S6 Fig. Subgroup meta-analysis by study setting of the association between individual-level water contact and odds of schistosome infection in children.** Full references of all included studies are available in S1 Table.
(TIFF)

**S7 Fig. Subgroup meta-analysis of the association between individual-level water contact and odds of schistosome infection in community-based studies.** Full references of all included studies are available in S1 Table.
(TIFF)

**S8 Fig. Funnel plots for meta-analysis of the association between daily, weekly, monthly water contact and odds of schistosome infection.** Egger's test p-value reported for all categories with N≥10.
(TIFF)

**S9 Fig. Influence analysis of any single study in the meta-analysis of the association between daily, weekly, monthly water contact and odds of schistosome infection.** Full references of all included studies are available in S1 Table.
(TIFF)

**S10 Fig. Funnel plot for meta-analysis of the association between individual-level water contact duration and odds of schistosome infection.** No Egger's test was conducted due to the small number of studies (N = 6) because Egger's test may lack statistical power to detect bias when N<10.
(TIFF)

**S11 Fig. Funnel plot for meta-analysis of the associations between water contact activities and odds of schistosome infection.** Egger's test indicates significant funnel plot asymmetry (p<0.001). The funnel plot includes effect sizes and standard errors from all water contact activities reported across studies, using a multilevel model clustered by study.
(TIFF)

**S12 Fig. Influence analysis of any single study in the meta-analysis of the associations between water contact activities and odds of schistosome infection.** Full references of all included studies are available in S1 Table.
(TIFF)

**S1 Table. Study characteristics and full references.**
(DOCX)

**S2 Table. Full list of excluded and included studies.**
(XLSX)

**S3 Table. Overview of final grouping of the exposure categories: having any water contact, water contact frequency water contact duration and water contact activities.**
(DOCX)

**S4 Table. Quality appraisal results.**
(DOCX)

**S1 Text. PRISMA checklist.**
(DOCX)

**S2 Text. Search strategy.**
(DOCX)

**S3 Text. Notes on data extraction.**
(DOCX)

**S4 Text. Quality appraisal tool.**
(DOCX)

## Acknowledgments

We thank Dr Reem Malouf and Nia Roberts for their support and guidance during the initial stages of the systematic review. We thank the authors of included studies who responded to our emails and provided additional data to reconstruct odds ratios.

## Author Contributions

**Conceptualization:** Fabian Reitzug, Goylette F. Chami.

**Data curation:** Fabian Reitzug, Julia Ledien.

**Formal analysis:** Fabian Reitzug.

**Funding acquisition:** Goylette F. Chami.

**Investigation:** Fabian Reitzug.

**Methodology:** Fabian Reitzug, Goylette F. Chami.

**Project administration:** Goylette F. Chami.

**Resources:** Goylette F. Chami.

**Software:** Goylette F. Chami.

**Supervision:** Goylette F. Chami.

**Validation:** Fabian Reitzug, Julia Ledien.

**Visualization:** Fabian Reitzug.

**Writing – original draft:** Fabian Reitzug.

**Writing – review & editing:** Fabian Reitzug, Julia Ledien, Goylette F. Chami.

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
