## [Decision Letter · Decision Letter 0]

24 Apr 2023

Dear Dr Chami,

Thank you very much for submitting your manuscript "Associations of water contact frequency, duration, and activities with schistosome infection risk: A systematic review and meta-analysis" for consideration at PLOS Neglected Tropical Diseases. As with all papers reviewed by the journal, your manuscript was reviewed by members of the editorial board and by several independent reviewers. The reviewers appreciated the attention to an important topic. Based on the reviews, we are likely to accept this manuscript for publication, providing that you modify the manuscript according to the review recommendations. 

Sincerely,

Matthew C Freeman, MPH, Ph.D.

Academic Editor

Francesca Tamarozzi

Section Editor

Reviewer's Responses to Questions

**Key Review Criteria Required for Acceptance?**

**Methods**

-Are the objectives of the study clearly articulated with a clear testable hypothesis stated?

-Is the study design appropriate to address the stated objectives?

-Is the population clearly described and appropriate for the hypothesis being tested?

-Is the sample size sufficient to ensure adequate power to address the hypothesis being tested?

-Were correct statistical analysis used to support conclusions?

-Are there concerns about ethical or regulatory requirements being met?

Reviewer #1: The methods are well explained and robust.

Reviewer #2: see attached comments

**Results**

-Does the analysis presented match the analysis plan?

-Are the results clearly and completely presented?

-Are the figures (Tables, Images) of sufficient quality for clarity?

Reviewer #1: The results have been analysed appropriately and presented clearly.

Reviewer #2: The analysis is clear, figures and tables are clear

**Conclusions**

-Are the conclusions supported by the data presented?

-Are the limitations of analysis clearly described?

-Do the authors discuss how these data can be helpful to advance our understanding of the topic under study?

-Is public health relevance addressed?

Reviewer #1: The conclusions are supported by the data presented. Just one suggested minor addition is that more could be said in the Discussion section about the need to re-define 'WASH' when we talk about schistosomiasis prevention, both in terms of the types of 'WASH' infrastructure that are needed and the level of community access and coverage that must be achieved. Typically 'WASH' refers to clean drinking water, toilets and handsoap. Indeed, universal access and usage to these components of WASH would eventually have a significant impact on schistosomiasis prevalence. However, the results of the study suggest that even small amounts of water contact can maintain infection levels, meaning that, at least in the near term, there is an urgent need to provide water infrastructure to facilitate safe water contact activities and replace the usage of contaminated water bodies for such activities. Examples may include safe laundry basins, showering/bathing facilities, or swimming facilities. It should be emphasised that when we say 'WASH' for schistosomiasis prevention, the 'W' must include enough safe water for all of a community's water contact needs, not just for drinking water. This is an under-emphasised point and means that even when communities are provided with traditional 'WASH' at sub-universal access levels, there is likely to still be significant schistosomiasis transmission risk.

Reviewer #2: The conclusion is based on the data presented

**Editorial and Data Presentation Modifications?**

Reviewer #1: There are no additional modifications needed.

Reviewer #2: See attached comments

**Summary and General Comments**

Reviewer #1: See my comment in the 'Conclusions' box above. Other than that, this was a well-executed study and this is an important paper for the NTD/schistosomiasis and WASH communities.

Reviewer #2: Overall the paper is clear and present a complex topic with many questions than answers.

PLOS authors have the option to publish the peer review history of their article (what does this mean?). If published, this will include your full peer review and any attached files.

Reviewer #1: No

Reviewer #2: No

Figure Files:

Data Requirements:

Reproducibility:

References

---

## [Editor Report · Decision Letter 1]

12 May 2023

Dear Dr Chami,

We are pleased to inform you that your manuscript 'Associations of water contact frequency, duration, and activities with schistosome infection risk: A systematic review and meta-analysis' has been provisionally accepted for publication in PLOS Neglected Tropical Diseases.

Best regards,

Matthew C Freeman, MPH, Ph.D.

Academic Editor

Francesca Tamarozzi

Section Editor

---

## [Editor Report · Acceptance letter]

1 Jun 2023

Dear Dr Chami,

We are delighted to inform you that your manuscript, "Associations of water contact frequency, duration, and activities with schistosome infection risk: A systematic review and meta-analysis," has been formally accepted for publication in PLOS Neglected Tropical Diseases.

Best regards,

Shaden Kamhawi

co-Editor-in-Chief

Paul Brindley

co-Editor-in-Chief
